# Study of the Process of Destruction of Harmful Microorganisms in Water

Askar Abdykadyrov [1], Sunggat Marxuly [1,*], Ainur Kuttybayeva [1,*], Nurgul Almuratova [2], Muratbek Yermekbayev [1,2], Serikbek Ibekeyev [1], Assel Yerzhan [2] and Yessen Bagdollauly [1]

1    Department of Electronics, Telecommunications and Space Technologie, Institute of Automation and Information Technologies, Satbayev University, Almaty 050000, Kazakhstan
2    Department of Electric Drive and Automation of Industrial installations, Institute of Electric Power and Electrical Engineering, Almaty University of Power Engineering and Telecommunications named after G.Daukeev, Almaty 050000, Kazakhstan
*    Correspondence: sungat50@gmail.com (S.M.); ainur_k_75@mail.ru (A.K.); Tel.:+7-702-327-80-15

**Abstract:** In this scientific work, the problem of studying the process of destruction of microorganisms in water by an Etro-03 device based on electric corona discharge is considered. In the research, a special Etro-03 ozonator device was developed for clearing water of biological pollutants. Testing of the installation was carried out in order to disinfect surface water in the Kapshagai reservoir. During the research, various harmful microorganisms were found in the composition of the primary water that did not meet the maximum permissible concentration (MPC). For example, coliphages, coli-indices, and the number of microbes in general came across in large numbers. During the technological process, various amounts of ozone ($O_3$) were released into the water, the amount and effective economic indicators of which were determined. In the same way, the effective time of the decontamination process was determined. During the research process, an algorithm of theoretical calculations was developed, and a mathematical model was given to bring $1m^3$ of surface water as the indicator for which sanitary rules and norms are approved.

**Keywords:** electric discharge; ozone; primary water; reservoir; ozonator; ozonated water; ozone content; decontamination

## 1. Introduction

Decontamination of surface water through an ozonator is more effective in comparison with methods of decontamination with chlorine or ultraviolet rays. First of all, ozone is more effective than any other disinfectant treatment in killing viruses and bacteria contained in water. In addition, it requires very little contact time. Furthermore, it reduces the decontamination time and does not leave chemical residues in the water in a short time. Due to the high oxidation potential of ozone in water, it effectively reduces germs and viruses. It causes rupture of the cell membrane and the breakdown of important biomolecular components in bacteria. Ozone can be used to oxidize the hydrocarbons of two layers of cellular lipids to kill the polluting microbes in water. During the technological process, there is practically no harmful waste with the use of ozone since ozone is subject to natural decomposition in water.

Ozone treatment of surface water also prevents the growth of harmful substances that were removed during the decontamination process. During the process, ozone is produced on-site and does not require transportation or processing, thus eliminating the difficulties associated with chemical treatment, such as safety and environmental issues. That is, in the process of water treatment, ozone is an effective disinfectant. The decontamination efficiency is measured by a mutually multiplied value of concentration and time. The degree of efficiency and reliability of ozone disinfection of water is primarily affected by the resistance of pathogenic organisms, as well as the concentration of ozone on them. In

recent years, there has been a rapid increase in intestinal viral infections spread mainly by water around the world (hepatitis A, gastroenteritis B), the incidence of polio, etc. Therefore, along with the quality of drinking water in terms of organoleptic and chemical indicators, issues of epidemiological safety are currently very relevant. In particular, they are considered in publications in a number of scientific journals. It shows the effectiveness of various disinfectant reagents.

Improvements to drinking water disinfection methods are currently developing in the following main areas:

- The application of ozonation technology;
- The use of ultraviolet (UV) rays;
- Radiation with accelerated electrons;
- The effect of high-frequency currents;
- The use of membranes, etc.

However, in the process of surface water disinfection in production, a real alternative to the use of chlorine is ozonation and water treatment with UV rays. The advantage of both methods is the prevention of the formation of organochlorine by-products as well as the high disinfection efficiency, including the removal of polio virus. The use of combinations of these methods has become quite widespread in Europe and America [1,2].

## 2. Materials and Methods

The allotropic form of oxygen generated by an electric discharge through ozone is currently widely used in production, including in agriculture, vegetables and fruits, dairy products, and agricultural devices as well as in water management and medicine. Due to the excessive use of pesticides and nitrogen fertilizers in agriculture, soil water, surface water, and groundwater are subject to chemical pollution. In order to neutralize such pollutants, an ozonator based on atmospheric oxygen was installed on the barrier discharge and disinfection works were carried out. In this work, fundamental research was carried out on the change in the acidity and amount of nitrogenous nutrients, bacteria, and soil DNA remaining in the soil after ozone treatment with a system using a quartz chamber [3,4]. In general, any device, be it soil or liquid, will require a certain amount of ozone to neutralize them, and ozone is produced in production with the help of ozonators of various designs. During the process of ozone production, its inefficiency increases if the energy consumption exceeds the required value. The problem of improving the methods of processing air-mixed gas at the ozonator outlet requires increasing the physical characteristics of the flow discharge and electrode tip, i.e., electron temperature and electron density. Some foreign research works used an emICCD chamber to study the characteristics of a flow discharge by observing the process of spreading the discharge in the head of a conical electrode with a positive voltage. Then, a flow discharge with a positive voltage and Thomson scattering at the electrode tip were performed. Thomson found that scattering is considered the most reliable method for simultaneously measuring the temperature and density of an electron in plasma [5]. At the same time, disinfection of drinking water based on electric field discharge and hydrodynamic cavitation gives positive results [6].

Today, as part of the normalization of the prevention and control of the epidemic in general, it is advisable to use traditional disinfection methods correctly. Disinfection devices can be used, for example, in plasma medicine. Plasma contains various highly active components that can effectively destroy pathogenic microorganisms. The advantage of the research work will be the waste-free, safe, and environmentally friendly technology. The dielectric barrier can be pre-decontaminated by a discharge-based device. At the same time, in this work, water activated by plasma can directly interact with microorganisms contained in it. That is, it was used in this work as the main technology for the study of its efficiency, environmental friendliness, and potential as equipment for convenient disinfection. In the same way, research on epidemic prevention and control was given a scientific justification and definition [7]. Although the determination of effective reactive oxygen species (rot) generated by plasma has been extensively studied, the mechanisms of

microbial disinfection and plasma disinfection processes have never been clearly explained. However, the research reveals in terms of comprehensive factors, including cell morphology, membrane permeability, lipid peroxidation, membrane potential, and intracellular redox homeostasis (intracellular rot and $H_2O_2$ and antioxidant system), the subclass mechanism of inactivation of yeast cells during plasma–liquid interaction [8].

In general, in the process of surface water disinfection, it is necessary to use environmentally friendly materials that achieve the inhibition of bacterial growth, aimed at improving water quality. In some scientific research work, silver nanoparticles (AgNPs) were developed to neutralize water in a warehouse by filtration. First, AgNPs were synthesized by the green synthesis method using an extract of Aloe Barbadensis Miller (Aloe Barbadensis Miller) as a reducing agent and $AgNO_3$ as a metal precursor. Silver nanoparticles are a platform for the production of multifunctional polymer membranes for wastewater disinfection [9].

Graphene oxide, a carbon-based nanomaterial, has an antibacterial effect in the composition of water due to its large surface area, surface charge density, and various physical and chemical properties. In one study, using the composition of graphene oxide/bismuth (GO/BiVO 4), disinfection of water containing *Escherichia coli* (*E. coli*) was performed. Nanoparticles of bismuth vanadate (BiVO 4) synthesized using the sol-gel method with a particle size of 21.3 nm were used to decorate graphene oxide (GO) sheets. Composites were developed consisting of five different combinations of 0.5%, 1%, 1.5%, 2%, and 2.5% GO/BiVO 4 (GO mass). The antibacterial ability of all five synthesized composites was tested in a homemade photoreactor controlled by visible light. The results showed the efficiency of the 1.5% GO/BiVO 4 nanocomposite in 0.1 g/L with a decontamination rate of 90% in 30 min under visible light radiation [10].

The purpose of this scientific research work is to discuss selected viruses, including COVID-19, and trends in their possible destruction in hot sources. Research in Colombia is limited, and this scientific discussion provides the necessary information regarding these viruses. This includes research on the following topics, both in Colombia and internationally:

(1)    Presence of viruses in hot springs;
(2)    Non-oxidizing water treatment in hot springs;
(3)    Virus removal strategies;
(4)    Impact of COVID-19 on hot springs.

The results of this discussion indicate that a monitoring scheme is needed to guarantee the elimination of the COVID-19 virus, to control it, and to achieve effective disinfection of water [11]. In the same way, the formation and control of bromate can be carried out through the process of biologically activating carbon by ozonation [12].

During the process of destroying microorganisms contained in water, ozone also breaks down some organic compounds. For example, 2-methylisoborneol and geosmin cannot be easily removed by conventional water procedures. The ozone and biologically activated carbon filter (O3-BAC) trend is the only option for removing methyl isoborneol and geosmin. For methyl isoborneol and geosmin with an initial concentration of 500 ng/L, the removal efficiency by ozonation and adsorption of activated carbon was 51–73%, 19–17%, and 30–10%. Accordingly, when the ozone content was 2.0 mg/L at this point, the contact time was 12 min (filter height 550 mm). Having a filter height of 300 mm was sufficient to lower methyl isoborneol and geosmin below the odorous threshold concentration at 500 ng/L. It can be seen that by ozonation and adsorption with activated carbon, methyl isoborneol (geosmin) is removed by about 48% to 57%. Similarly, nitrofuran antibiotics in water can be broken down by ozonation [13,14]. Additionally, in the mode of continuous water purification, zinc hydroxide heterogeneous catalytic ozonation works also showed good results [15].

Furthermore, the heterogeneous catalytic ozonation of P-chloronitrobenzene was promoted by glass-supported zinc hydroxide in water through the use of a water continuous flow mode.

Due to the high concentration of nitrogen and phosphorus in the composition of purified water, algae formation occurs over time. When ozonating such waters, it is possible to destroy algae but not completely destroy the cell. Although ozone cannot directly remove algae and some nutrients from the water composition, it can contribute well to the biodegradation of water. With the process of effective decontamination of such waters, the ozone content will be 5–10 mg/L [16].

The process of ozonation of drinking water is one of the most important steps for treatment plants. The efficiency of ozonation directly depends on the quality of the purified water. In this work, an extended control scheme based on observation of the model forecast for the process of ozone dosing is proposed. With the proposed control scheme, a model of the radial main function neural network was created, which can be used to control the model forecast. For effective disinfection requirements, a control strategy of maintaining permanent exposure to ozone is adopted. The results of a full-scale experiment demonstrate the effectiveness of this advanced control method [17]. Similarly, during the technological process, it is necessary to take into account the mass transfer system in the ozone contactor for drinking water purification [18].

Although pre-coagulation treatment of water with ozone is widely used in foreign research work, the effect of pre-ozonation on coagulation to remove particles and natural organic matter from micro-contaminated surface water is not yet clear. A pilot study was carried out on the effect of pre-ozonation before the coagulation of micro-contaminated surface waters of the Queshan reservoir from the Yellow River. Effective results were shown with the amount of 3.0 mg/L of ozone for the destruction of micro-contaminated river water particles and natural organic matter. In addition, the effectiveness of the pre-ozonation process was determined (the efficiency of removing blur, color, UV 254, and COD Mn increased by 24–35%) [19].

Various experiments have been used to treat water according to the characteristics of reservoir water contaminated with micro-pollutants. In addition, when using pre-oxidation with ozone, it can be seen that large-sized particles are reduced in the same way that organic substances with unsaturated chemical bonds are clearly reduced. In the general research, it was observed that ozone can be effectively treated with preliminary use—that is, in combination with traditional technologies—to eliminate turbidity, organic matter, and coliform bacteria [20]. At the same time, the combination of ozonation and ion exchange processes for urban wastewater treatment gave the best results in the study [21].

Ozone had a good effect on the decomposition of dichloroacetic acid in a nano-sized ZnO solution in water [22]. In addition, the results of the analysis of the liquid by chromatography in the work on 2,4-D and its ozonation are presented [23].

## 3. Result

When ozone is introduced into the composition of water, it reacts to various components in the water, including microorganisms, as well as the processes of oxidation of heavy and light metals in the chemical composition and decomposition of organic compounds such as fats. Water destroys microbacteria, oxidizing the chemical elements they contain. Ozonation of water has the following advantages over chlorination:

- Ozone not only affects the redox system of bacteria but also directly affects the protoplasm.
- As a result of the high oxidizing potential of ozone, the bactericidal effect in water is stronger than that of other chemical agents.
- Ozone acts 15–20 times faster than chlorine. For example, if polio virus is killed in 2 min (ozone content 0.45 mg/L), chlorine at a dose of 2 mg/L kills it after 3 h.
- The required amount of ozone is about 2.5 times less than that of chlorine.

### 3.1. Ozone Decomposition during the Technological Process

The process of dissolving ozone in water is the basis of many scientific research works. The solubility of ozone in water is usually described by the Henry coefficient or a solubility

coefficient (Rt). In scientific experimental work, the solubility coefficient (effective Henry coefficient) depends on the given temperature:

$$R_t = C_c / C_\Gamma \tag{1}$$

where $C_c$ and $C_\Gamma$ are the concentrations of ozone in the liquid and gas phases, respectively.

These characteristics mainly include temperature, the pH of the medium, electrical conductivity, and the presence of impurities. Further interaction of the HO2 ion with ozone leads to the formation of radical ions. The tendency of ozone to dissipate in water can be traced in the following chemical bond expression [24]:

$$
\begin{aligned}
O_3 + OH^- &\rightarrow HO_2^- + O_2 \\
O_3 + HO_2^- &\rightarrow O_2^- + OH \\
O_3 + O_2^- &\rightarrow O_3^- + O_2 \\
O_3^- + H_2O &\rightarrow OH + OH^- + O_2
\end{aligned}
\tag{2}
$$

Currently, in some scientific research works, the possibility of the participation of oxygen atoms in the process of ozone decomposition in acidic solutions is being considered [25]. The decay of the oxygen atom of ozone, formed as a result of heat, is the basis of the process.

$$O_3 + M \rightarrow O_3{}^* \rightarrow O + O_2 + M \tag{3}$$

where $O_3{}^*$ is oscillatory, excitable ozone and M is a molecule which is considered for the gas phase and as a monomolecular process [26].

The occurrence of such a reaction in water is confirmed by the inhibition of the process. With an increase in oxygen content in ozone–oxygen mixtures, it is better decomposed in aqueous solutions [24].

Thus, the solubility coefficients of ozone in water were determined by ozone (Rt), with a pH of 0.1–11. A decrease in Rt is associated with the decay of ozone, the mechanism of which is direct in nature. The greatest stability of ozone solutions is shown in a strongly acidic environment, where the catalytic effect of hydroxyl ions is significantly reduced and protonation occurs (highly active radical ions are inactive).

### 3.2. Practical Testing of the Device

Taking into account the above-mentioned properties of ozone, a pilot ozonator Etro-03 based on electric corona discharge was specially developed at the Department of Electronics, Telecommunications, and Space Technologies of the Kazakh National Research Technical University, named after K. I. Satbayev. The general scheme of the installation is presented in Figure 1 below.

In order to study the process of destruction of harmful microorganisms in water, a technological scheme of water disinfection was developed based on a special Etro-03 pilot ozonator device (Figure 2).

In the process of ozone disinfection of water, the amount of ozone (0.5–5 mg/L) consumed depends on the content of the contaminated water. Purified water contains a residual ozone content of 0.1–0.4 mg/L. Thus, ozonation is one of the modern methods of water purification that is, to some extent, universal. This is because ozone simultaneously has bacteriological organoleptic and chemical effects on pollutants in water. The bacterial effect of ozone depends on the pH value of the water (pH = 6–10 and water temperature range from 0 to 37 °C). The effectiveness of the bactericidal effect of ozone is influenced by the color of the suspended, dissolved organic substances and the presence of chemical additives in small quantities. The presence of floating solids in treated water somewhat limits the possibilities of ozone because, in most cases, suspended particles are protectors of bacteria. That is, they protect them by absorbing them on the surface. There is no clear decision on the negative effect of dissolved organic substances on the bactericidal effect. Some scientific researchers believe that the disinfecting effect of ozone is manifested only at a certain residual (or excess) concentration in water, when dissolved organic pollutants

are oxidized. Many foreign scientific research papers indicate that the disinfecting effect of ozone is observed simultaneously with the oxidation of organic substances. It is very difficult to assume that ozone has a selective effect; first it acts on dissolved organic impurities, and then on bacteria [27]. The activity of the most common disinfectants is shown in Table 1.

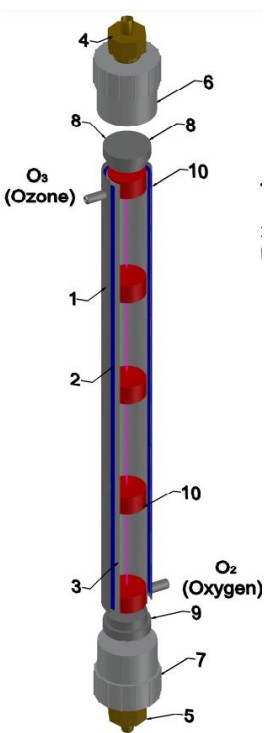

**Figure 1.** Ozonator installation "Etro-03" based on electric corona discharge: 1—dielectric tube; 2—$NO_2$ or $H_2O$ transformer oil travel space; 3—non-oxidizing tube; 4, 5—cover made of brass; 6, 7—cover made of dielectric material; 8, 9—cover made of fluoroplastic material; 10—sealing material stabilizing gate.

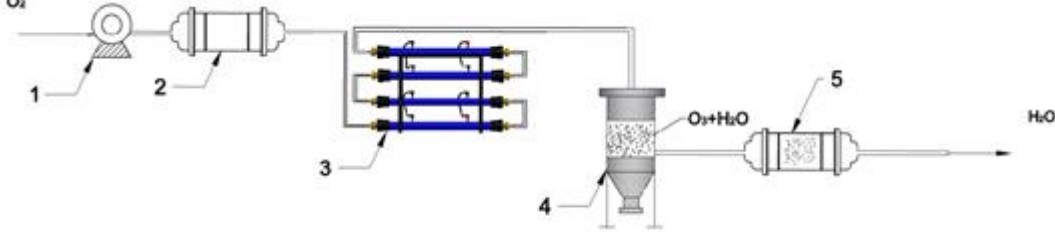

**Figure 2.** Technological process of destruction of harmful microorganisms in water: 1—compressor; 2—air drying device; 3—ozonator unit Etro-03; 4—water decontamination tank; 5—membrane filter.

**Table 1.** Activity of the most common disinfectants.

| Disinfectant Methods | Bacteria (*Escherichia coli*) | Poliovirus 1 | Cysts (*Entameba histolytica*) |
|---|---|---|---|
| Ozone ($O_3$) | 1950 | 840 | 2.7 |
| Chlortecic acid (HOCl) | 110 | 1.7 | 0.12 |
| Chlorine dioxide ($ClO_2$) | 10 | 1.21 | - |
| Hypochlorite ions $OCl^-$ | 3.5 | 0.12 | - |
| Dichloramine ($NHCl_2$) | 0.52 | 0.0011 | - |
| Chloramine ($NH_2Cl$) | 0.013 | 0.0012 | - |

The efficiency of ozonation was higher compared to other disinfection methods, as can be seen from Figure 3 below, for example [28]. During the research, various disinfection methods (temperature = 20 °C; pH = 7) were used against indicator bacteria (the effectiveness of cleaning *E. coli* reached up to 99%) and viruses.

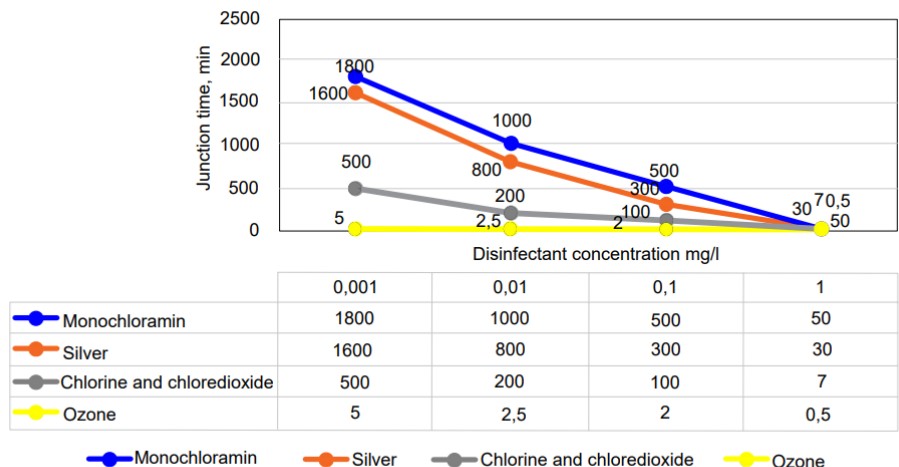

**Figure 3.** The effect of various disinfectant methods on *E. coli* bacteria (where T = 280c and pH = 7.5).

To determine the effectiveness of disinfectant oxidants in water supply systems, it was necessary to develop a sanitary reliability criterion, taking into account the types and doses of various reagents, the length of the water supply network, and quality indicators. The introduction of such a criterion into practice was included in the basic drinking water supply law in the United States in 1986 [29].

This criterion is relevant for the dose of disinfectants (mg/L) and the product of contact time (min). Its value is not the same for different oxidants and "objects" of action. Table 2 below shows the criterion values required for 99% decontamination.

**Table 2.** Value criteria.

| Microorganisms | Ozone pH 6-7 | Chlorine pH 6-7 | Chloramine pH 8-9 | Chlorine dioxide pH 6-7 |
|---|---|---|---|---|
| *E. coli* | 0.018 | 0.028 | 87 | 0.5 |
| Polyviruses | 0.11 | 1.1 | 650 | 0.3 |
| Rotaviruses | 0.005 | 0.01 | 3730 | 0.2 |
| Giardia cysts | 0.54 | 18 | up to 2100 | 25 |
| Guardia muris cysts | 1.7 | 55 | up to 1250 | 6.8 |
| Cryptosporidia | 1.8 | up to 7100 | up to 6800 | up to 65 |

When disinfecting drinking water with ozone, the criterion value is usually taken equal to 1.6 (taking into account the preservation of the residual ozone concentration of 0.4 mg/L within 4 min). The role of water temperature during the decontamination process is very important for any oxidants.

In their work, Coena et al. found that a residual ozone level of 0.4 mg/L for 4–6 min is sufficient to destroy polyviruses. In addition, in this work, the concept of the "C × T-criterion", called Watson's law, was introduced:

$$\log(N_t : N_0) = -K \times C \times t \tag{4}$$

where $N_t$ and $N_0$ represent the concentration of microflora initially (0) and at a certain time (t), C is the concentration of the decontaminating reagent; and K is the constant of microflora reactivation.

In the practice of decontamination and purification of surface water, the concept of marginal concentration or inactivation ability of ozone in relation to viruses and bacteria is introduced. The maximum concentration of ozone in water is 0.4 mg/L. This residual ozone concentration destroys 99.99% of microorganisms in 4 min in the contact chamber. The main disadvantage of ozone during the technological process is that, due to the lack of conservative efficiency, some microorganisms reappear in the water after decontamination.

Taking into account the remoteness of water consumer zones, it is advisable to use additional reagents so that bacteria do not reappear in the water after ozonation. Since ozone in general is a powerful disinfectant, its residual concentration in water disappears quickly (especially due to increased pH and temperature). The results of the research work can be seen in Figure 4 below.

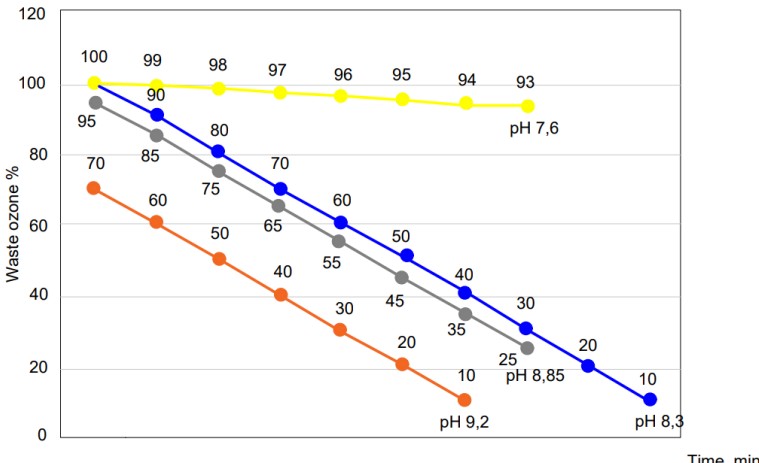

**Figure 4.** Decomposition of ozone in water depending on the pH.

In water that has passed the entire complex of classical treatment plants, including water disinfection, there is an increase in the activity of bacteria and an increase in their number after the decomposition of ozone. It has been observed that under the influence of ozone, the amount of biodegradable compounds increases as a result of the destruction of organic matter in the water. For the same reason, it promotes the regrowth of microorganisms in the water supply network. Therefore, when transporting water over long distances, it would be reasonable to disinfect with reagents containing ozone and additional chlorine (chlorine, chloramines, or chlorine dioxide).

In order to conduct research testing the ozonator plant, special water was removed from the Kapshagai surface reservoir and ozonation was carried out. The microbiological indicators of the water content do not correspond to the amount of MPC, as can be seen in Table 3 below.

**Table 3.** Microbiological indicators of water content in the reservoir.

| | Microbiological Indicators of Water Content | Maximum Permissible Concentration (MPC) | Microbiological Indicators of Primary Water Content Number | Ozone Content in g/h | | | |
|---|---|---|---|---|---|---|---|
| | | | | 0.3 | 0.4 | 0.5 | 0.6 |
| 1 | Total number of microbes, 1 mL (colony-forming units) | <50 | 150 | 130 | 120 | 40 | 15 |
| 2 | Thermotolerant coliform bacteria (100 mL CFU) | 0 | 350 | 200 | 100 | 50 | 0 |
| 3 | Common coliform bacteria (100 mL CFU) | 0 | 700 | 440 | 260 | 112 | 0 |
| 4 | Coliphages (100 mL CFU) | 0 | 150 | 70 | 30 | 12 | 0 |

For example, in the table, the total number of microbes should not exceed (50) per MPC. In our case, it can be seen that the initial water content was 150, and the amounts of TCB (100 mL of CFU), CCB (100 mL of CFU), and coliphages (100 mL of CFU) did not reach the MPC at all. When releasing different amounts of ozone into water, the microbiological indicators in it gradually decrease, as can be seen in Figures 5 and 6 below. In Figure 5, we can observe changes in the total number of microbes and the mutual influence of ozone. We noted that with an ozone content of 0.5–0.6 g/h, the total number of microbes in the water satisfied the fixed condition in the MPC.

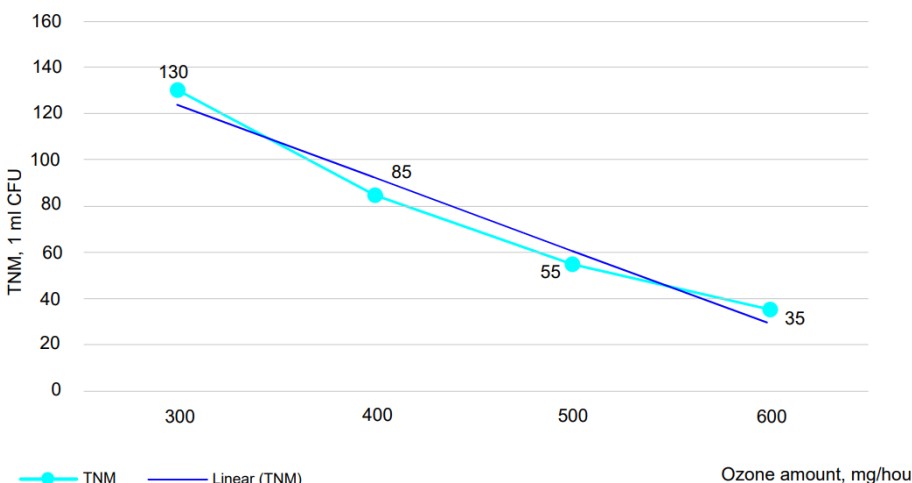

**Figure 5.** Interactions between the total number of microbes and ozone dependence.

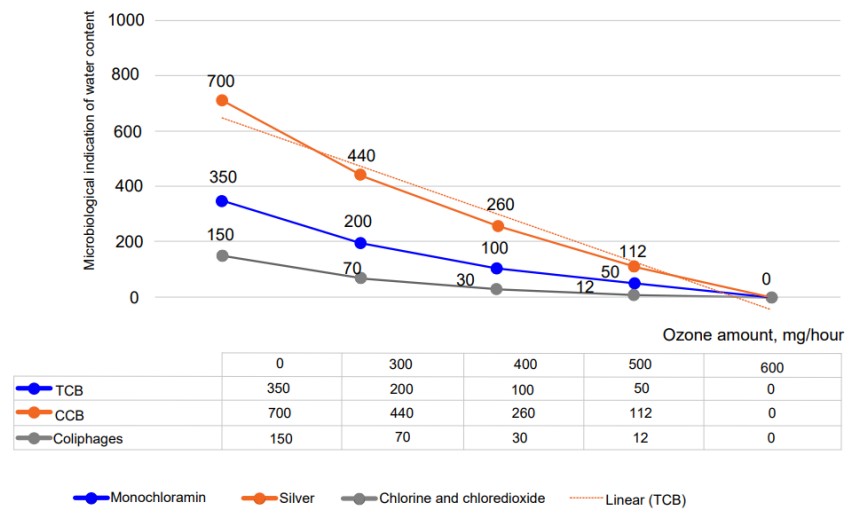

**Figure 6.** Interaction of water composition with microbiological indicators and ozone dependence.

In Figure 6, we can see changes in the amounts of TCB (100 mL of KOE), CCB (100 mL of KOE), and coliphages (100 mL of BOE) and the interaction of ozone. We noted that at an ozone content of 0.6 g/h, microbiological pollutants contained in the water reached the MPC. However, in order to bring experimental work into a single system—that is, to determine the amount of ozone consumed per 1 $m^3$ of water—it is advisable to build a mathematical model taking into account physical parameters [30].

### 3.3. Mathematical Model of the Technological Process

To create a mathematical model of the process of destroying harmful microorganisms in water, work was carried out to calculate some parameters using a special SMath Solver program [31]. The algorithm of the research work is presented in Figure 7 below. As

shown in the figure, theoretical calculations were carried out following an algorithm for reducing and eliminating excess water content (100 mL of KOE), OCD (100 mL of KOE), and coliphages (100 mL of BOE) according to the technological process. Theoretical calculations were considered according to two options.

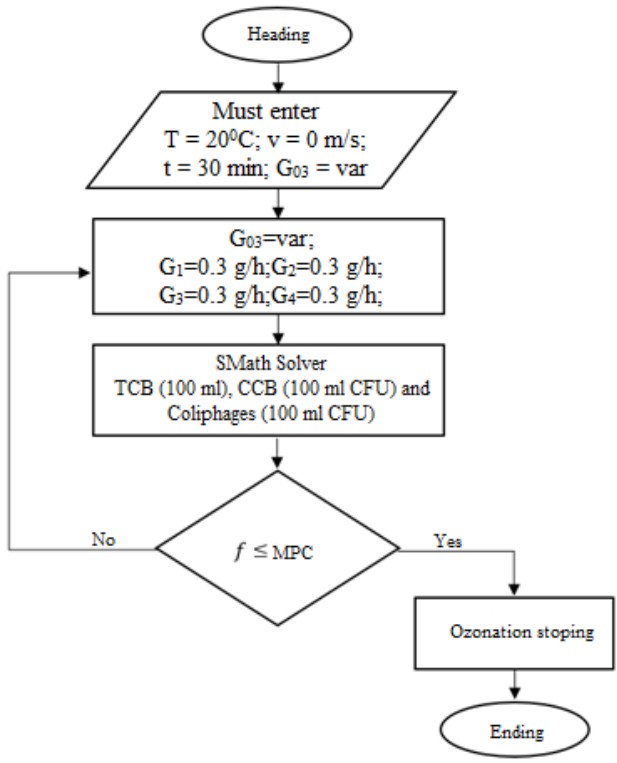

**Figure 7.** Algorithm of the process of destruction of harmful microorganisms in water.

The first option: According to the first version, during the process, the concentration of ozone was changed while keeping the decontamination time constant ($t$ = const) at $t$ = 0.5 h = 30 min. The maximum permissible concentration (MPC) can be calculated as follows:

$$f = \frac{1}{K \cdot G \cdot t}, \ (100 \ \text{mL CFU}) \tag{5}$$

where $K$ is the amount of ozonated bacteria in water of different quantities, $G$ is the amount of ozone (G/H), and $t$ is the decontamination time (minutes).

(4) On changing the concentration of ozone ($G_{ozone}$, g/h) using the expression, the MPC value can be calculated as follows:

$$f_1 = \frac{1}{K \cdot G_1 \cdot t} = 0.0071 (100 \ \text{mL CFU}) \tag{6}$$

$$f_2 = \frac{1}{K \cdot G_2 \cdot t} = 0.003 (100 \ \text{mL CFU}) \tag{7}$$

$$f_3 = \frac{1}{K \cdot G_3 \cdot t} = 0.0015 (100 \ \text{mL CFU}) \tag{8}$$

$$f_4 = \frac{1}{K \cdot G_4 \cdot t} = 0.0009 \ (100 \ \text{mL CFU}) \tag{9}$$

Figure 7, representing Equation (5), shows that for the process of destruction of harmful microorganisms in water by the algorithm, the effective amount of ozone is $G_{ozone}$ = 0.6 g/h. During the technological process, it is possible to neutralize the composition of water from harmful compounds by changing the time constant at some point. If we keep

the amount of ozone in the water constant (Gozone = const.) and change the decontamination time, we can determine the effective time constant.

The second option: According to this version, it is possible to observe a decrease in the number of microbes in general, including harmful microbiological indicators in water, even when changing the time and keeping the corresponding ozone concentration (Gozone = const.) constant. The mathematical calculations of the research work in theoretical terms can be traced in Equation (7), and its algorithm can be seen in Figure 8 below.

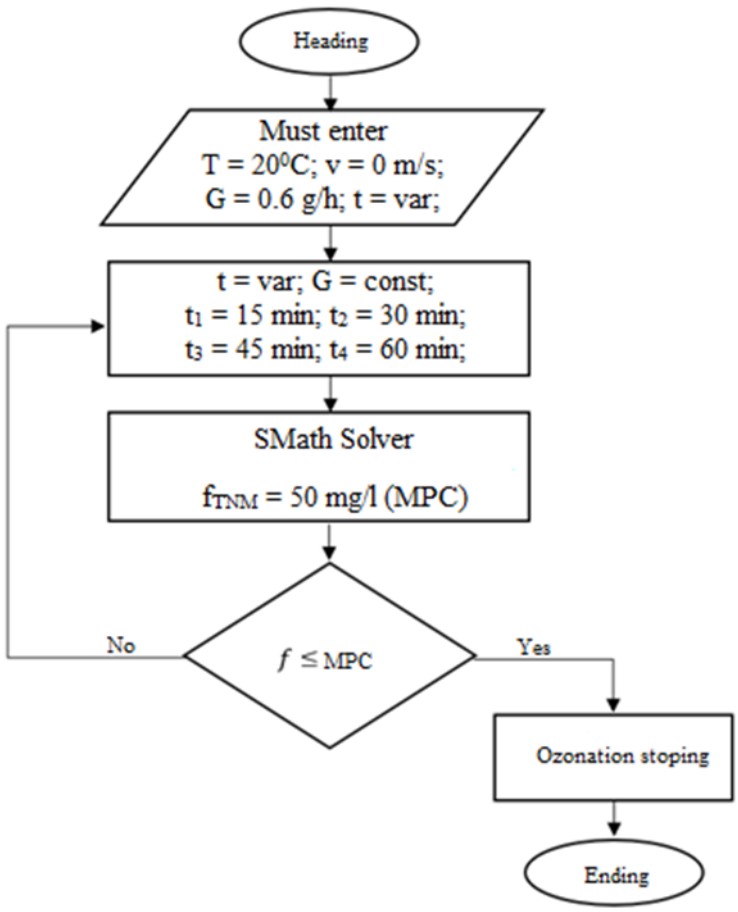

**Figure 8.** Algorithm of the process of destruction of harmful microorganisms in water (Gozone = const.). Must enter: T = 200 C; v = 0 m/s; G = 0.6 g/h; t = var.

In the study, the decrease in the total number of microbes in the water when changing the disinfection time of the tank water can be observed in Table 4 below.

**Table 4.** Microbiological indicators of reservoir water content (Gozone = const).

| № | Microbiological Indicators of Water Content | Maximum Permissible Concentration (MPC) | Microbiological Indicators of Primary Water Content Number | Ozone Content 0.6 g/h | | | |
|---|---|---|---|---|---|---|---|
| | | | | $t_1 = 15$ min | $t_2 = 30$ min | $t_3 = 45$ min | $t_4 = 60$ min |
| 1 | Total number of microbes, 1 mL of CFU | <50 | 130 | 0.05 | 0.02 | 0.017 | 0.012 |

As can be seen in Figure 9 and Table 4, in the first 15 min, the total number of microbes in the water reached the MPC. In the quality indicator of drinking water, it is necessary not to exceed the requirements established as sanitary rules and norms (the total number of microbes in it is 50).

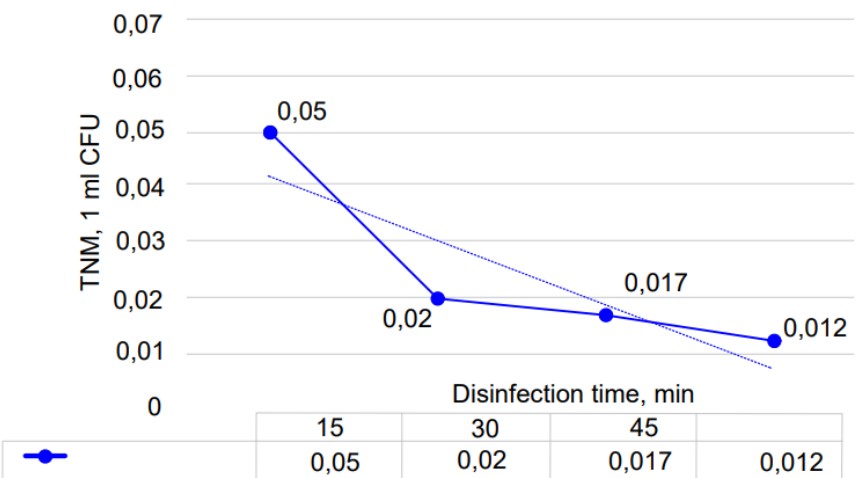

**Figure 9.** Interaction of water composition with microbiological indicators and ozone time.

Figure 8 shows decontamination—that is, the longer the contact time, the greater the quality of the water. The experimental data presented in the figure can theoretically be calculated with T = 200 C, v = 0.00 m/s, G = 0.60 g/h, and t = var. Additionally, t1 = 15 min, t2 = 30 min, t3 = 45 min, t4 = 60 min, and fTNM = 50 mg/l (according to MPC). Depending on the time elapsed during the decontamination process, the total number of microbes can be determined as follows (TNM—total number of microbes).

$$f_1 = \frac{1}{K \cdot G \cdot t_1} = \frac{1}{130 \cdot 0.6 \cdot 0.25} = 0.0513 \ (1 \text{ mL of CFU}) \tag{10}$$

$$f_2 = \frac{1}{K \cdot G \cdot t_2} = \frac{1}{130 \cdot 0.6 \cdot 0.5} = 0.0256 \ (1 \text{ mL of CFU}) \tag{11}$$

$$f_3 = \frac{1}{K \cdot G \cdot t_3} = \frac{1}{130 \cdot 0.6 \cdot 0.75} = 0.0171 \ (1 \text{ mL of CFU}) \tag{12}$$

$$f_4 = \frac{1}{K \cdot G \cdot t_4} = \frac{1}{130 \cdot 0.6 \cdot 1} = 0.0128 \ (1 \text{ mL of CFU}) \tag{13}$$

If the total number of microbes in the water F ≤ MPC does not meet the condition, then the decontamination process will have to be extended. In the standard approved by sanitary standards and regulations, the total number of microbes (1 mL of CFU) should not exceed <50. During the research work, it can be observed that in the first 15 min, f = 0.0513 (1 mL of CFU) is equal to the MPC.

## 4. Discussions

In order to study the process of destruction of harmful microorganisms in water, scientific research on water disinfection—that is, the elimination of pathogenic microorganisms, viruses, and bacteria from water—was discussed. The technological process is based on redox reactions that are triggered when adding a disinfectant component to water or using reagent-free methods. Unlike sterilization, the process does not aim to completely destroy germs. Methods of water disinfection—that is, the preliminary choice of disinfectant technology—depend on the types and amount of microorganisms contained in the water. When using reagents, it is important to create conditions in which the oxidizer can penetrate the microorganism and destroy its DNA (deoxyribonucleic acid) and RNA (ribonucleic acid) proteins. This is necessary in order for them to stop the reproduction of microorganisms. Microbes are located differently in water, so they are resistant to chemical disinfection.

The residual concentration product value (mg/dm$^3$) and contact time (min) of disinfectants are used to compare the efficiency in killing bacteria and different viruses contained

in water. As part of the research work, a review of the literature was carried out, where the advantages of various disinfectants and chemical oxidants were considered:

- Oxygen oxidizers;
- Chlorine and chloroform reagents;
- Ozone;
- Other neutralizing oxidants.

We focused on the properties of disinfectants with the help of UV radiation and much more. The effectiveness of all of the above oxidizers can be seen in Figure 10 below.

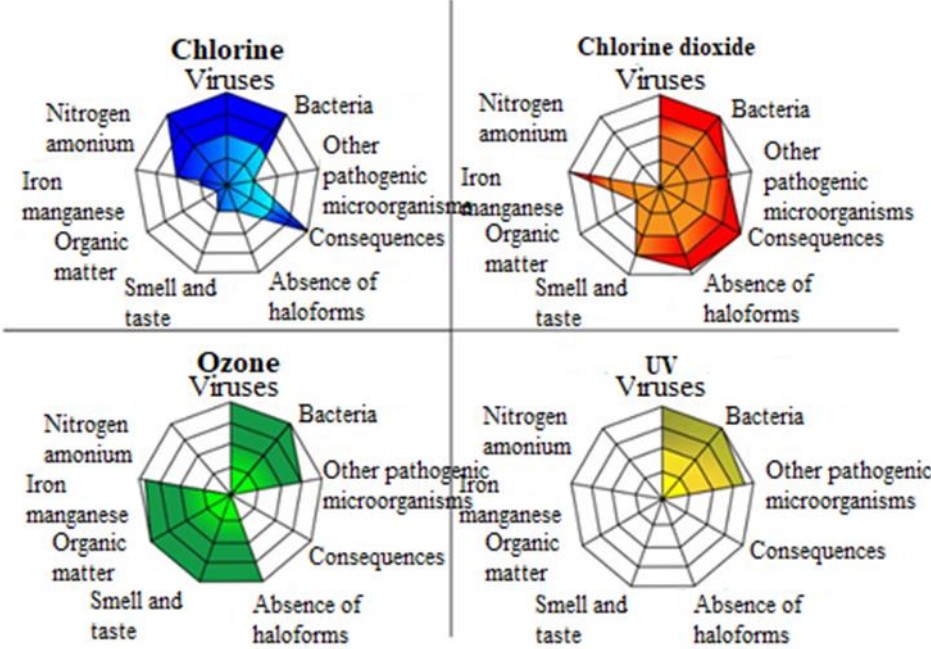

**Figure 10.** Effectiveness of disinfectants in the process of surface water disinfection.

According to the figure, a difference between ozone and other oxidizing reagents can be observed. For example, viruses, bacteria, and the smell and taste of water can also be seen as highly effectively dealt with in the process of oxidizing heavy metals. To this end, in the research work, we considered the main methods of decontamination of drinking and industrial waters. At the same time, the technological scheme of the process of ozonation of surface water by means of a pilot ozonator was studied. However, in production, water reaches the consumer through many kilometers of pipes, and along the way, it can be subjected to secondary pollution. Such water can be disinfected with a high-quality water filter. It was noticed in a research paper that the most effective filter is a membrane filter, which allows one to obtain water that can be consumed as-is without harm to health.

## 5. Conclusions

In conclusion, a literature review of various scientific research works was carried out in order to study the process of eliminating harmful microorganisms contained in water. Based on the results of the discussion of the research works, the specifics of the process of decontamination and purification of surface water using ozone technology can be seen in Figure 10. The direction of this research on the process of destroying harmful microorganisms contained in surface water was based on ozone technology. Here, a small new type of pilot ozonator, Etro-03, was developed at the Department of Electronics, Telecommunications, and Space Technologies of the Kazakh National Research Technical University, named after K.I. Satbayev, for the disinfection of contaminated natural water in reservoirs using ozone technology. For practical testing of the plant, the following results

were revealed when performing ozonation work with water extracted from the Kapshagai reservoir:

- Microbiological indicators in the water showed the complete disappearance of thermotolerant coliform bacteria, coliform bacteria in general, and microbes found in water in general.
- It was found that organoleptic indicators in the composition of the water changed the smell, taste, and precipitate of the water, and some heavy metals (iron and copper) were oxidized, precipitated, or turned into some other chemical compound.
- There was a decrease in toxicological indicators found in the water, namely inorganic compounds with a high toxicity indicator such as nitrates, nitrites, fluorine, etc.
- Toxicological indicators found in the composition of water were observed, in which there was a decrease in organic compounds.

The above research was carried out directly with the pilot ozonator installation Etro-03, based on the electric corona discharge created in the new model. The research was carried out in the period from 2015 to 2022, and the main technological indicators of the installation were determined.

**Author Contributions:** Conceptualization, A.A. and S.M.; methodology, A.A. and S.M.; software, A.K. and S.M.; validation, S.I., M.Y. and Y.B.; formal analysis, A.A. and S.M.; investigation, A.A., N.A., M.Y. and S.M.; resources, A.A. and S.M.; data curation, A.A. and S.M.; writing—original draft preparation, A.A. and S.M.; writing—review and editing, A.A. and S.M.; visualization, A.A. and S.M.; supervision, A.A. and S.M.; project administration, A.A. and S.M.; funding acquisition, A.A., S.M., A.K., N.A., S.I., A.Y., M.Y. and Y.B. All authors have read and agreed to the published version of the manuscript.

**Funding:** This research received no external funding.

**Institutional Review Board Statement:** Not applicable.

**Informed Consent Statement:** Not applicable.

**Data Availability Statement:** Not applicable.

**Conflicts of Interest:** The authors declare no conflict of interest.

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
