# Peer review of "Study of the Process of Destruction of Harmful Microorganisms in Water"

_water, doi:10.3390/w15030503_

Round 1
Reviewer 1 Report
The manuscript entitled "Study of the process of destruction of harmful microorganisms in water" is not a novel study even though the work is not explained well. The figures are taken as a screenshot and I think they are unsuitable for publication. In my opinion, the paper can be accepted for publication after major revision.
Author Response
fixed

Reviewer 2 Report
There are several issues on this manuscript. They are
(i) Provide research gap with clear objectives.
(ii) There are several errors in English. On line 48, etc.therefore,
(iii) On line 58, at the use of membranes, etc.м. What does M. stand for ?
(iv) On lines 259-260, the water temperature ranges from 0 - 370c. The degree centigrade should change into correct form.
(v) On line 274, temperature 20 C.The degree centigrade should change into correct form.
(vi) On Table 1, there are so many commas. For examples, 3,5 and 0,52 etc. Recheck.
(vii) Figure 3. The effect of various disinfectant methods on E. Сoli bacteria 284 (where t = 280c pH = 7.5). Here, I think "t" stands for temperature and it should be T instead of t. So, t should be time.
(viii) not completed caption of Table 2. Value criteria? What does Value criteria stand for ?
(xi) On Table 2, there are so many commas. For examples, 1,7 and 1,8 etc. Recheck.
(x) On line 294, coena et al. should be Coena et al.
(xi) On line 327, For example, in the table: which table number?
(xii) The manuscript looks like a report. Correction is needed.
(xiii) Rewrite the conclusions by including exact findings from results and discussion.
Author Response
fixed

Reviewer 3 Report
In the scientific work submitted for review, the problem of studying the process of destruction of microorganisms in water by an Etro - 03 device based on electric corona discharge is considered. In the research work, a special Etro-03 ozonator device was developed for cleaning the water composition from biological pollutants. Testing of the installation was carried out in order to disinfect surface water in the Kapshagai reservoir.
This is very interesting research, providing the basis for the implementation of further projects, however, the article requires corrections:
1. please correct line 58; what does M after etc mean?
2. please remove the dot from the title of chapter 3
3. please use subscripts in chemical formulas in all cases, and in the case of ions - superscripts
4. in lines 260 and 274, 285 write down the temperature values and units correctly
5. in line 275 it is not enough to write E.
6. please write Latin names of microorganisms in italics
7. in table 1 you need to complete the units
8. the drawings are poorly readable, their quality needs to be improved, this applies to all figures
9. please standardize the notation of units throughout the text, I suggest, for example: mg/L and not mg/l; you can't write mg/hour and mg/h or g/h interchangeably, it needs to be corrected, unified
10. line 332 is missing a space
11. please correct the captions under figures 6, 8, 9
12. in the article we have two formulas with the number 4 and after correcting it, the numbering of the others will change, this change also requires correction in the text
13. in line 398 there was a font change
14. please correct chapter 4
15. please justify the text in the discussion section
16. please correct the entry in line 456 and the year in 465
17. please correct the bibliography in accordance with the guidelines for authors; in my opinion citations in Cyrillic are not legible for everyone.
The publication is a valuable source of information to implement new solutions in this field and forms the basis for further research.
Thank you for considering my opinion. I encourage the authors to continue working on improving the manuscript.
Author Response
fixed

Round 2
Reviewer 1 Report
The appropriate revisions have been made and accepted for publication.
Reviewer 2 Report
The revised manuscript is fine.